# Diagnostic value and complication rate of ultrasound-guided transthoracic core needle biopsy in mediastinal lesions

Rosen Petkov[1], Tzvetan Minchev[2], Yordanka Yamakova[3], Evgeni Mekov[1]*, Georgi Yankov[1], Danail Petrov[1]

1 Department of Pulmonary Diseases, MHATPD 'Sveta Sofia', Medical University – Sofia, Sofia, Bulgaria, 2 Thoracic Surgery Department, Acibadem Tokuda Hospital, Sofia, Bulgaria, 3 Department of Anesthesiology and Intensive Care, National Oncology Hospital, Medical University – Sofia, Sofia, Bulgaria

* dr_mekov@abv.bg

## Abstract

### Background

Ultrasound-guided transthoracic core needle biopsy (US-TCNB) is a promising method for establishing the correct diagnosis of mediastinal masses. However, the existing studies in this area are scant and with small samples.

### Purpose

To evaluate the diagnostic value and the complication rate of US-TCNB, particularly large bore cutting biopsy in patients with mediastinal lesions.

### Material and methods

This retrospective study includes 566 patients with mediastinal lesions suspicious of malignancy evaluated between March 2004 and December 2018. Inclusion criteria: 1. Patients with mediastinal lesions detected on thoracic CT scan; 2. Lesions more than 15 mm; 3. Negative histological diagnosis after bronchoscopic biopsy; 4. Normal coagulation status; 5. Cooperative patient; 6. Written informed consent. US visualization of the mediastinal lesions was successful in 308 (54.4%). In all of them, US-TCNB was performed. All patients with mediastinal lesions unsuitable for US visualization were evaluated for a CT-guided transthoracic needle biopsy (CT-TTNB), which was done if the presence of a safe trajectory was available (n = 41, 7.2%). All patients inappropriate for image-guided TTNB were referred to primary surgical diagnostic procedures (n = 217, 38.3%).

### Results

The US-TCNB is a highly effective (accuracy 96%, sensitivity 95%) and safe tool (2.6% complications) in the diagnosis of all subgroups mediastinal lesions. It is non-inferior to CT-TTNB (90%) and comes close to the effectiveness of surgical biopsy techniques (98.4%), but is less invasive and with a lower complication rate.

**Data Availability Statement:** The datasets are available at Figshare: https://figshare.com/articles/Med_TU17_CT_sav/12044346 for CT biopsies;

https://figshare.com/articles/Mediastinal_US-TCNB/12044340 for US biopsies.

**Funding:** The author(s) received no specific funding for this work.

**Competing interests:** The authors have declared that no competing interests exist.

## Conclusion

US-TCNB of mediastinal lesions is highly effective and safe tool which is particularly helpful in critically ill patients.

## Introduction

The morbidity of mediastinal tumors increases in the last 15 years and in 2014 it is 0.5/100,000 in men and 0.4/100,000 in women [1]. The main imaging modalities used in the evaluation of mediastinal abnormalities (chest X-ray, CT, MRI, and PET) have high sensitivity in the detection of mediastinal lesions and could give valuable information about the size, location and metabolic activity [2]. However, imaging and clinical evaluation often do not allow a conclusion regarding the final diagnosis in these patients. Therefore, cyto-histopathologic diagnosis is often required.

Percutaneous needle biopsy of mediastinal lesions under fluoroscopic guidance was first described in 1967 [3]. Since 1980 computed tomography had almost replaced fluoroscopy as a guide for a transthoracic needle biopsy (TTNB) [3]. CT guidance has the advantage of accurate localization of the lesion and reducing the risk of puncturing vessels in the mediastinum [4].

On the other hand, ultrasound (US) guidance has advantages such as real-time monitoring of the biopsy; the possibility of oblique needle paths and the ability to perform a biopsy in critically ill patients. However, the existing studies in this area are scant and with small samples.

This study aims to evaluate the diagnostic value and the complication rate of ultrasound-guided transthoracic core needle biopsy (US-TCNB), particularly large bore cutting biopsy, in patients with mediastinal lesions.

## Materials and methods

This retrospective study includes patients evaluated between March 2004 and December 2018.

Inclusion criteria: 1. Patients with mediastinal lesions detected on thoracic CT scan; 2. Lesions more than 15 mm; 3. Patients without a histological diagnosis after bronchoscopic biopsy and transbronchial fine-needle aspiration (FNA); 4. Normal coagulation status; 5. Cooperative patient.

Exclusion criteria: 1. Uncorrectable abnormalities in coagulation: INR>1.4; platelets <50000/ml; aPTT >1.5 times above the referent range.

The decision tree illustrating the diagnostic approaches to the patients included in the study is given in Fig 1. All patients underwent preliminary CT and US examination to localize the target lesion and to identify a safe path for needle placement. US-TCNB under real-time monitoring was carried out in all patients with mediastinal lesions suitable for US visualization. We used US system with 2D, color, Angio Doppler and contrast-enhanced imaging modalities, equipped with a sector (2.0–2.5 MHz), convex (3.5-5-7.5) MHz and linear (5–7.5–12.0 MHz) transducers, B-mode, Color, Angio and pulse wave Doppler Mode options. As acoustic windows, we used parasternal and paravertebral intercostal spaces, upper thoracic aperture and transdiaphragmal access. B-mode and color Doppler ultrasound accurately localized the tumor, clearly depicted its anatomical relations to the mediastinal vessels, the margin of the lung and the location of the internal mammary artery. The depth, location, internal structure, and size of the lesion were evaluated in B-mode images. The vascularity of the lesion was detected by color Doppler sonography.

Different US-guided biopsy techniques were used, as follow: (a) Attachable stretcher guides that can be fit to the transducers, directing the needle to various depths from the transducer,

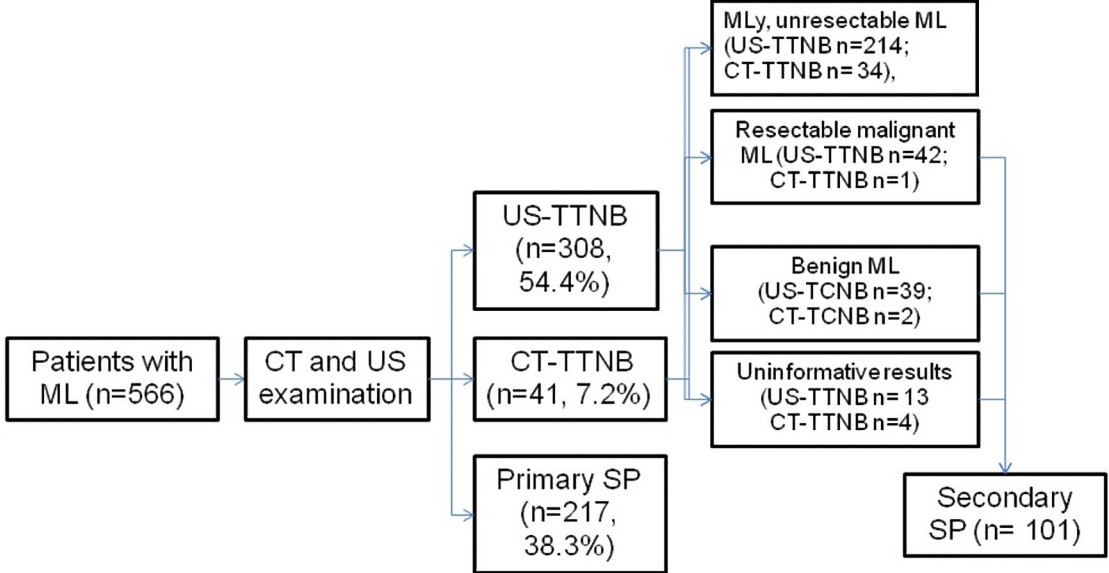

**Fig 1. The decision tree illustrating the diagnostic approaches to the patients included in the study.** Abbreviations: ML: mediastinal lesions; MLy: malignant lymphoma; CT: computed tomography; US: ultrasound; US-TCNB: US-guided transthoracic needle biopsy; CT-TTNB: CT guided transthoracic needle biopsy; SP: surgical procedures.

depending on the angle set by the operator; (b) "Free hand" technique: the needle can be inserted through the skin directly into the plane of view of the transducer. During the US-TCNB the echogenic needle tip was guided to the target by B-mode sonography and a 'twinkling sign' on color Doppler imaging [5] (Figs 2 and 3).

In the first 125 patients, US-TCNB was preceded by ultrasound-guided transthoracic fine-needle aspiration (US-TFNA) for evaluation of diagnostic value between the methods. US-TFNA was carried out using a 22 G Chiba needle.

The parasternal approach was used for US-TCNB biopsy in patients with anterior-superior mediastinal lesions. The same approach may be utilized in rare cases with lesions in middle

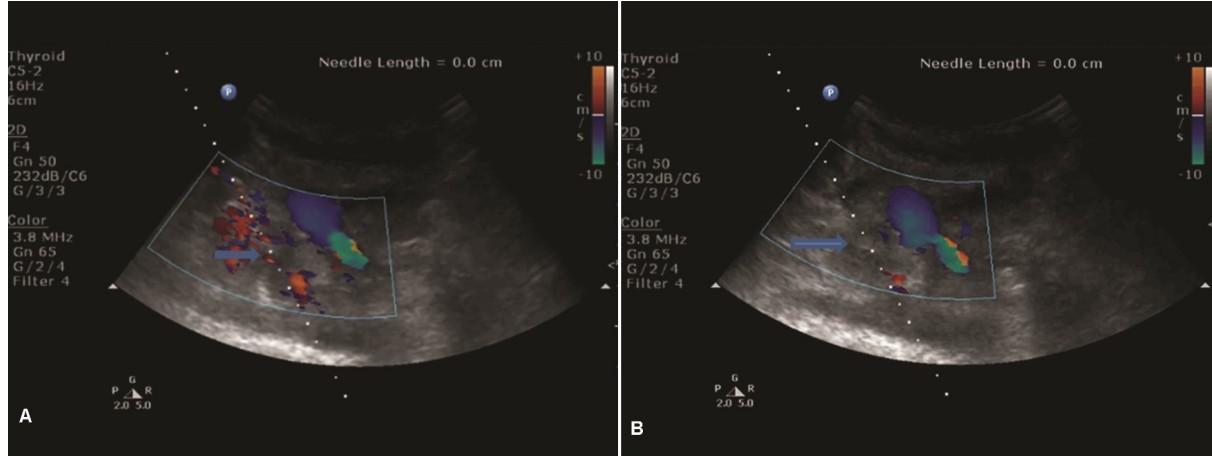

**Fig 2.** A. Color Dopler US imaging: a "twinkling sign" with visualization of the biopsy needle during its manual movement (arrow). B. Color Dopler US-imaging supraclavicular approach, convex probe 3.5 MHz: Enlarged upper paratracheal lymph nodes (R2). US-TCNB (18G): The biopsy needle placed into the target mediastinal lesion (arrow).

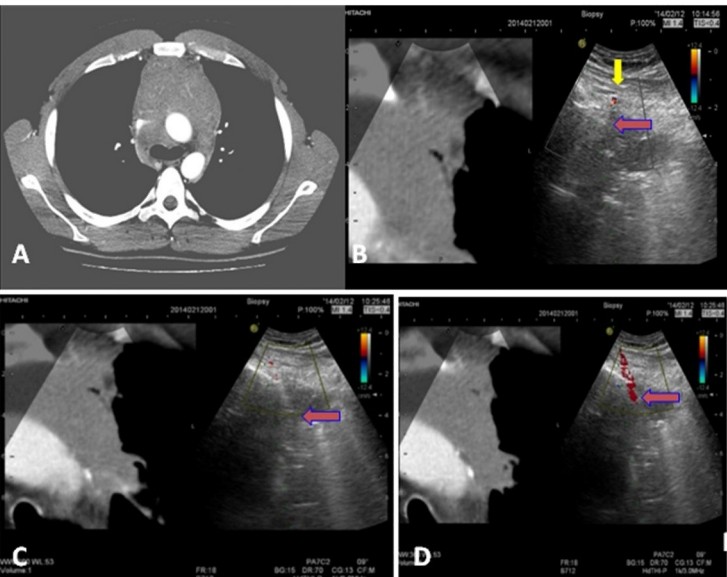

**Fig 3.** A. CT scan: Large mediastinal lesion in the anterior-superior compartment. B. Real-time Virtual Sonography (RVS) fusion imaging technology CT/ US Color Doppler (CD) imaging, left site parasternal approach (transversal section), with a convex probe 2.5–5.0 MHz: Visualization of a. thoracica int. sin. (yellow arrow), the biopsy needle (18G) placed across the thoracic wall. C. Biopsy needle into the tumor mass (blue arrow marks the apex). D. Twinkling sign (CD phenomenon) marks the needle.

mediastinum surrounded by atelectatic pulmonary parenchyma. The internal mammary artery was evaluated by color Doppler sonography to avoid being affected during the biopsy (Fig 3). The internal mammary vessels located on either side of the sternum may have a significant variation in the position [6]. A suprasternal or supraclavicular approach through the paratracheal soft tissue space was used to detect lesions located in the upper mediastinum. If possible, the patients were in a supine position with a pillow below the scapulas to keep the neck maximally extended. The paravertebral approach was used for biopsy of posterior mediastinal lesions. All patients with mediastinal lesions unsuitable for US visualization were evaluated for a CT-guided TTNB (CT-TTNB). The CT guiding was done using a CT system General Electric Bright Speed, 0.75 s gantry rotation time, 100–120 kV, 50–80 mA, 5 mm collimation. In a parasternal approach, the needle is inserted lateral to the sternum and advanced through the chest wall and mediastinal fat into the target lesion. CT scans were obtained between steps of needle advancements to check the trajectory and ensure that the mediastinal vessels are not in the needle path. A direct mediastinal approach, which allows extrapleural needle placement, was the preferred method of US/CT-TTNB to avoid the risk of pneumothorax.

US/CT-TCNB was performed using large-bore (14G) or small-bore (18G) automatic histologic needles with a length of 100–200 mm. All image-guided TTNB were performed under local anesthesia, using a 10 ml 1% solution of Lidocaine. The patients were followed up at least 3 hours after the procedure to ensure their hemodynamic stability and to monitor their respiratory status. In all patients chest X-ray and control US examination were obtained 2 hours after the biopsy. US-TFNA samples were pathologically and microbiologically examined, including hematoxylin-eosin (HE), acid-fast, and Gram stains or bacterial cultures in case of clinical suspicion. Special histologic staining methods including immunohistochemical techniques were applied.

All patients inappropriate for image-guided TTNB were referred for primary surgical diagnostic procedures. The patients with benign mediastinal lesions, those with malignancies considered as resectable and patients with uninformative results according to US-TCNB and CT-TTNB were referred to secondary surgical procedures (Fig 1).

The definitive diagnosis was verified by histopathological examination of a tissue sample obtained through image-guided TCNB, primary and secondary surgical procedures. All results from the US/CT-TTNB techniques were subsequently reviewed and compared with the definitive diagnosis. Sensitivity, specificity, positive and negative predictive values and diagnostic accuracy of the method were estimated by standard formulas.

Statistical analysis was performed using commercially available software (Microsoft Excel; Microsoft Corp. and SPSS Statistic 17; SPSS Inc.). The data from the study was processed with methods of descriptive statistics. A chi-square test was used to determine the associations between categorical variables. Continuous variables were examined for normality by the Shapiro-Wilk test. For normally distributed variables, differences between the groups were determined by independent-samples T-test. Mann-Whitney U test was used for abnormally distributed variables. A P-value of 0.05 was considered significant.

The Clinical Center for Pulmonary Diseases Ethical Committee approved the study and all patients provided written informed consent for the procedure (number 48/16.08.2017). All the information is processed anonymously.

## Results

A total of 566 patients with mediastinal lesions above 15 mm in diameter (mean 55.4 ± 19 mm, minimum 15 mm, maximum 160 mm), suspicious of malignancy were evaluated, 306 (54%) were males and 260 (46%) were females, mean age 46.8 ± 16.8 years. The mediastinal lesions were localized according to the surgical division of the mediastinum into compartments (Fig 4) [5].

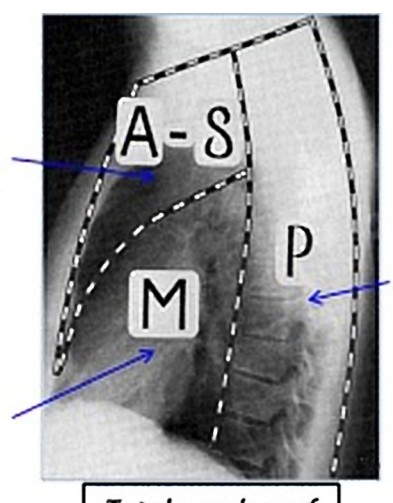

**Fig 4. Distribution of the patients according to the localization of mediastinal lesions.** Surgical division of the mediastinum into compartments: A-S—anterior-superior mediastinal compartment; M—Middle mediastinal compartment; P—posterior mediastinal compartment.

US visualization of the mediastinal lesions was successful in 308 (54.4%) patients, with the highest success ratio (84%) for the lesions located in the anterior-superior compartment (Fig 4). In all of them, US-TCNB was performed. The US guidance of TTNB allowed real-time continuous monitoring of the needle during advancement and sampling. The color Doppler US imaging allowed great vessels, collaterals and tumor vessels to be demonstrated clearly, and the biopsy route could be preselected to avoid puncturing them and as much as possible to prevent complications (Figs 2 and 3). The parasternal approach (Fig 3) was used for US-TCNB biopsy in 247 (80.2%) patients: 245 of them with anterior-superior mediastinal lesions and 2 patients with middle mediastinum lesions visualized through atelectatic lung parenchyma. There were 38 patients for whom US-TCNB through supraclavicular and suprasternal approach was considered safe (12.3%, Fig 2). The paravertebral approach was used for biopsy of posterior mediastinal lesions in 23 patients (7.5%). The distribution of patients according to the localization of the mediastinal lesions and the type of performed image-guided TTNB procedure is presented in Table 1.

We performed a single pass, large-bore (14G) US-TCNB in 228 patients. In 54 of them, we repeated the biopsy due to not good quality tissue obtained, and in 3 patients we had to do the procedure for the third time. A total of 288 14G US-TCNB were performed in 228 patients (1.26 procedures per patient).

Small-bore (18G), two passes US-TCNB were carried out in 80 patients. We did not establish a statistically significant difference in dimensions of mediastinal lesions biopsied with 14G and 18G true cut needles (Table 1; p>0.05).

At the time of presentation 113 (20%) patients had superior vena cava syndrome (SVC). We succeeded in ultrasound visualization of the target lesion in 85 (75.2%). In this high-risk group, a single pass 14G (n = 53) and two passes 18G (n = 32) US-TCNB were performed. In 22 critically ill patients with severe dyspnea who cannot tolerate a supine position, US guidance provided us with an opportunity to perform TCNB biopsy in sitting or semi-sitting positions.

US-TCNB (14G or 18G) in patients with mediastinal lesions provided an informative tissue sample, which allowed establishing a correct histological diagnosis in 95.8% (295/308) of them.

The final diagnoses according to the histological results from US-TCNB and confirmed by surgical procedures are listed in Table 2. In 1 patient the diagnosis of malignant thymoma, achieved by US-TCNB was subsequently revised following the histological result of a surgical biopsy (thoracotomy) in Hodgkin's lymphoma. In 13 patients (4.2%) US-TCNB/FNA specimens were uninformative and the diagnosis was established through: thoracotomy (n = 5); VATS (n = 7); US-TCNB of supraclavicular lymph node (n = 1). The definitive diagnoses

**Table 1. Distribution of the patients according to the localization of ML and the type of performed image-guided transthoracic needle biopsy.**

| Methods of biopsy | A-S comp. | Middle comp. | Posterior comp. | Dimensions Min/ Max $\bar{X} \pm$ SD | TTNB passes $\bar{X}$ |
|---|---|---|---|---|---|
| US-TCNB (n = 308) | 283 | 2 | 23 | 15 / 160 mm | |
| | 216 | 0 | 12 | 57 ± 20 mm | 1.26 |
| 14G (n = 228) | 67 | 2 | 11 | 52 ± 20 mm | 2.0 |
| 18G (n = 80) | 114 | 1 | 10 | 52.5 ± 15 mm | 2.0 |
| US-FNA (n = 125) | | | | (p>0.05) | |
| CT-TTNB (18G) (n = 31) | 15 | | 16 | 20 / 85 mm | 1.1 |
| CT-FNA (n = 10) | | 10 | | 50± 13 mm (p>0.05) | 2.0 |

Abbreviations: TTNB: transthoracic needle biopsy; US-TCNB: US-guided transthoracic core needle biopsy; US-FNA: US-guided transthoracic fine-needle aspiration; CT-TTNB: CT-guided transthoracic core needle biopsy; CT-FNA: CT-guided fine-needle aspiration; A-S comp.: anterior-superior mediastinal compartment.

**Table 2. Histological diagnosis of mediastinal lesions according to the results of US-TCNB, confirmed by surgical procedures (n = 308).**

| Diagnosis | Number of patients | Percent (%) |
|---|---|---|
| Hodgkin's lymphoma | 62 | 20.1 |
| Thymoma | 43 | 14.0 |
| Benign mesenchymal tumors | 35 | 11.4 |
| Diffuse large B-cell lymphoma | 33 | 10.7 |
| Metastasis from SCLC | 33 | 10.7 |
| Metastasis from NSCLC | 26 | 8.4 |
| Other types of non-Hodgkin lymphomas | 21 | 6.8 |
| Malignant mesenchymal tumors | 18 | 5.8 |
| Metastasis from extrathoracic tumors | 9 | 2.9 |
| Malignant germ cell tumor | 5 | 1.6 |
| Carcinoid tumor | 4 | 1.3 |
| Mediastinal teratoma | 2 | 0.6 |
| Substernal thyroid goiter | 2 | 0.6 |
| Malignant neurogenic tumor | 1 | 0.3 |
| Intrathoracic thyroid gland carcinoma | 1 | 0.3 |
| Uninformative result | 13 | 4.2 |
| Total | 308 | 100% |

were: Hodgkin's lymphoma (n = 5); large B-cell lymphoma (n = 3); lymphoblastic lymphoma (n = 1); malignant thymoma (n = 2); malignant mesenchymal tumor (n = 1); metastatic small cell lung cancer (n = 1).

The accuracy of all US-TCNB for the diagnosis of malignant mediastinal lesions was 95.8% (295/308) (Table 3). The diagnostic value of US-TCNB in special clinical conditions and by type of the mediastinal lesion is given in Table 3. There is no significant difference in accuracy between 14G and 18G US-TCNB. We did not find a reliable difference in the accuracy of the US-TCNB between subgroups patients with and without SVC and/or mediastinal syndrome (94.1% vs. 96.4%, p>0.25).

**Table 3. Diagnostic value of all minimally invasive methods.**

| Method | Acc, % | Se, % | Sp, % | PPV, % | NPV, % |
|---|---|---|---|---|---|
| US-TCNB (n = 308) | 95.8 | 95.1 | 100 | 100 | 75 |
| - 14G true cut (n = 228) | 95.6 | 94.9 | 100 | 100 | 75.6 |
| - 18G true cut (n = 80) | 96.5 | 95.8 | 100 | 100 | 72.7 |
| US-TCNB in special clinical conditions and types of the mediastinal lesion | | | | | |
| SVC syndrome (n = 85) | 94.1 | 94 | 100 | 100 | 28 |
| Malignant lymphoma (n = 124) | 97 | 93 | 100 | 100 | 95.3 |
| Metastatic ML (n = 66) | 99.7 | 98.5 | 100 | 100 | 99.6 |
| Malignant thymoma (n = 36) | 99 | 95.6 | 99.6 | 97.7 | 99.2 |
| US-FNA (n = 125) | 76 | 73 | 94 | 98 | 37 |
| CT-TTNB (n = 41) | 90 | 90 | 100 | 100 | 33 |
| Image-guided (US and CT) TCNB (n = 339) | 95.3 | 94.6 | 100 | 100 | 72 |

Abbreviations: TTNB—transthoracic needle biopsy; US-TCNB—US-guided transthoracic core needle biopsy; US-FNA—US-guided transthoracic fine-needle aspiration; CT-TTNB—CT-guided transthoracic needle biopsy; n- number of patients; SVC—superior vena cava syndrome; ML mediastinal lesions. Acc—accuracy; Se —sensitivity; Sp—specificity; PPV—positive and NPV—negative predictive value.

There was no significant difference in dimensions of mediastinal lesions biopsied under US and CT control (Table 1; p = 0.23). The accuracy of CT-TTNB was 90% (Table 3). We did not establish a significant difference in accuracy between groups patients with US-TCNB and CT-TTNB.

The cytology examination of the US-TFNA specimen established a correct diagnosis in 78 patients (62.4%). In our study US-TFNA had high diagnostic accuracy for epithelial metastatic disease, establishing a correct diagnosis in 94% (32/34). US-TFNA specimen was guiding to the diagnosis in 65% (32/49) of cases with malignant lymphoma and 58% (14/24) of cases with thymoma. However, the accuracy of US-TFNA was significantly lower compared to US-TCNB (76% vs. 95.8%, p<0.001; Table 3).

The diagnostic value of all types of primary and secondary surgical procedures performed in patients with mediastinal lesions is presented in Table 4. The accuracy of all surgical procedures (n = 271) was 98.4%, sensitivity 98.2%, specificity 100%, positive predictive value 100% and negative predictive value 88.6%. There was no significant difference in the accuracy of the US-TCNB (n = 308) when compared to all surgical procedures (n = 318) (95.8% vs. 98.4%, p>0.1), but there is significant difference in accuracy between all image-guided (US and CT, n = 339) TCNB and all surgical procedures (p<0.025).

We observed complications of US-TCNB in 8 (2.6%) patients: 2 cases of partial pneumothorax, 1 of them needed pleural drainage with 3-day hospitalization; 3 patients with chest wall hematoma spontaneously resolved, 2 with collapses due to vasovagal reactions, 1 with non-massive hemoptysis. The complication rate did not depend on the size of biopsy needles: 14G (5/228) or 18G (3/80), p>0.10.

We found out complications risk of CT-TTNB 14.6% (6 patients): 3 cases of partial pneumothorax, 1 of them needed pleural drainage with 3-day hospitalization; 1 patient with a small amount of hemoptysis; 2 patients with collapses due to vasovagal reactions. The complication rate of US-TCNB was significantly lower than the complication rate of CT-TCNB (p<0.001).

## Discussion

Since *Saito* et al. showed 82% diagnostic yield of US-TCNB in a small sample of 11 patients, there is growing interest in this procedure [7]. *Sawhney* et al. described the use of large cutting needles for mediastinal biopsies in selected patients [8].

In our study, US-TCNB was an appropriate diagnostic method in 54% of the patients with mediastinal lesions, especially when lesions were located in the anterior-superior mediastinal compartment (success rate of US visualization 84%). We did not establish a significant

**Table 4. Distribution of the patients according to the performed surgical procedures.**

| Surgical procedures | VATS | TT | AMT | MSc | Sum SP |
|---|---|---|---|---|---|
| Primary SP | 169 | 25 | 2 | 21 | 217 |
| Secondary SP | 23 | 53 | 19 | 6 | 101 |
| Total number | 192 | 78 | 21 | 27 | 318 |
| Acc, % | 98.4 | 100 | 95.3 | 96.3 | 98.4 |
| Se, % | 98.2 | 100 | 95 | 96.2 | 98.2 |
| Sp, % | 100 | 100 | 100 | 100 | 100 |
| PPV, % | 100 | 100 | 100 | 100 | 100 |
| NPV, % | 87 | 100 | 50 | 50 | 88.6 |

Abbreviations: VATS—video-assisted thoracoscopic surgery; TT—thoracotomy; AMT—anterior mediastinotomy; MSc—mediastinoscopy; SP—surgical procedures.
Acc—accuracy; Se—sensitivity; Sp—specificity; PPV—positive and NPV—negative predictive value.

difference between the accuracy of US-TCNB and CT-TTNB. Our results correspond to the data of published studies in which the sensitivity of CT-TTNB varies between 70–94% with negative predictive value 35–55% [9,10] and complication rate 16–45% [11]. CT guidance has the advantage of accurate localization of the lesion, especially in cases invisible by US, thus the hazard of puncturing vessels in the mediastinum could be avoided [4]. However, CT-TTNB showed a higher incidence of iatrogenic complications when compared to US-TCNB. The additional disadvantages of CT-TTNB were higher cost, radiation exposure, and step by step control of the procedure.

The advantages of US guidance include: effective real-time monitoring of the biopsy; the color Doppler allows the vascularity of the lesion to be evaluated and vascular structures to be avoided as much as possible to prevent possible complications; the opportunity of changing needle slope during the biopsy procedure; the ability to perform a biopsy in critically ill patients including those with dyspnea in forced semi-sitting or sitting positions (not possible with CT guidance). In patients with SVC syndrome, a condition in which surgical diagnostic procedures carry a high risk of bleeding, respiratory distress, and airway compression, US-TCNB is highly effective and has the lowest complication rate [12–14].

The small-bore TCNB also provided high-quality histopathologic specimens adequate for histologic diagnosis in most cases. There is no significant difference in accuracy between subgroups with 14G and 18G US-TCNB. The US-TCNB was highly effective in verifying all diagnostic subgroups: malignant lymphoma, thymoma, metastatic mediastinal engagement. The complications of US-TCNB were mild, non-life threatening and did not depend on the size of the biopsy needle.

*Koegelenberg* et al. performed US-TFNA (n = 45) with rapid on-site evaluation followed by US-TCNB in undiagnosed cases [15]. This approach yielded similar diagnostic value: an accurate cytological diagnosis was made in 33 (73.3%), and was more likely to be diagnostic in epithelial carcinoma and tuberculosis (28/30) than all other pathologies (5/15). In our study, the accuracy of the US-FNA biopsy was comparable (76%). Despite advances in cytopathologic techniques, such as flow cytometry, immunohistochemical phenotyping, and gene rearrangement studies, the role of FNA in the diagnosis of lymphoma, thymoma, germ cell tumors, neurogenic tumors, and benign tumors remains controversial.

Our data suggest that US-TCNB should be a preferable method for histological verification of mediastinal lesions, especially for those located in the anterior-superior compartment. US-FNA biopsy is applied in high-risk patients alone or most often as an assistant method, tracing the safe path of the cutting needle to the target lesion.

Surgical procedures have accuracy up to 100% and in some cases could simultaneously establish the diagnosis and provide treatment [2,16–21]. All surgical procedures usually require hospitalization, general anesthesia and have a higher cost. Not least, complications are observed in 3–16% of patients [2,17–26].

To the best of our knowledge, this is by far the biggest study that examines the diagnostic yield and complication rate of US-TCNB in patients with mediastinal lesions. It shows the possibility of noninvasive histological diagnosis of mediastinal lesions while at the same time examines the complication rate. Moreover, it could be used in critically ill patients, including those with SVC syndrome.

The major limitation of US-TCNB was the absence of the 'acoustic window', observed in 46% of our patients.

## Conclusion

US-TCNB is a highly effective and safe diagnostic tool for mediastinal lesions in all nosological groups. According to the accuracy, it is non-inferior to the CT-guided needle biopsy and

comes close to the effectiveness of surgical biopsy techniques, but has significantly lower complication rate. It is the method of choice in critically ill patients, including those with SVC syndrome, especially when is necessary to perform a biopsy in forced semi-sitting or sitting position.

## Author Contributions

**Conceptualization:** Rosen Petkov, Evgeni Mekov.

**Data curation:** Rosen Petkov, Tzvetan Minchev, Yordanka Yamakova, Georgi Yankov, Danail Petrov.

**Formal analysis:** Rosen Petkov, Evgeni Mekov.

**Investigation:** Rosen Petkov, Tzvetan Minchev, Yordanka Yamakova, Georgi Yankov, Danail Petrov.

**Methodology:** Rosen Petkov.

**Project administration:** Rosen Petkov, Evgeni Mekov.

**Software:** Rosen Petkov, Evgeni Mekov.

**Supervision:** Rosen Petkov, Evgeni Mekov, Danail Petrov.

**Validation:** Rosen Petkov.

**Visualization:** Rosen Petkov, Evgeni Mekov.

**Writing – original draft:** Rosen Petkov, Yordanka Yamakova, Evgeni Mekov.

**Writing – review & editing:** Rosen Petkov, Tzvetan Minchev, Yordanka Yamakova, Evgeni Mekov, Georgi Yankov, Danail Petrov.

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
