## [Decision Letter · Decision Letter 0]

11 Dec 2019

PONE-D-19-21908

Diagnostic value and complication rate of ultrasound guided transthoracic core needle biopsy in mediastinal lesions

PLOS ONE

Dear Dr Mekov,

Thank you for submitting your manuscript to PLOS ONE. After careful consideration, we feel that it has merit but does not fully meet PLOS ONE’s publication criteria as it currently stands. Therefore, we invite you to submit a revised version of the manuscript that addresses the points raised during the review process.

We would appreciate receiving your revised manuscript by Jan 25 2020 11:59PM. To enhance the reproducibility of your results, we recommend that if applicable you deposit your laboratory protocols in protocols.io, where a protocol can be assigned its own identifier (DOI) such that it can be cited independently in the future. For instructions see: http://journals.plos.org/plosone/s/submission-guidelines#loc-laboratory-protocols

We look forward to receiving your revised manuscript.

Kind regards,

Azra Alizad, MD

Academic Editor

PLOS ONE

Journal Requirements:

Please ensure that your manuscript meets PLOS ONE's style requirements, including those for file naming. The PLOS ONE style templates can be found at http://www.plosone.org/attachments/PLOSOne_formatting_sample_main_body.pdf and http://www.plosone.org/attachments/PLOSOne_formatting_sample_title_authors_affiliations.pdfIn the ethics statement in the manuscript and in the online submission form, please provide additional information about the patient records used in your retrospective study. Specifically, please ensure that you have discussed whether all data were fully anonymized before you accessed them. If patients provided informed written consent to have data from their medical records used in research, please include this information. We noted in your submission details that a portion of your manuscript may have been presented or published elsewhere. ["Paper is containing original research and has not been submitted/published earlier in any journal and is not being considered for publication elsewhere. However there are 4 ABSTRACTS presented at the European Respiratory Congresses: ... They are not full text publications, nor this manuscript is submitted for consideration anywhere else. These are presentations of the work only in abstract form."] Please clarify whether this publication was peer-reviewed and formally published. If this work was previously peer-reviewed and published, in the cover letter please provide the reason that this work does not constitute dual publication and should be included in the current manuscript.We note that you have stated that you will provide repository information for your data at acceptance. Should your manuscript be accepted for publication, we will hold it until you provide the relevant accession numbers or DOIs necessary to access your data. If you wish to make changes to your Data Availability statement, please describe these changes in your cover letter and we will update your Data Availability statement to reflect the information you provide.

Additional Editor Comments (if provided):

Thank you for submitting your work to PLOS One. Please respond to the comments from the reviewer.

Reviewers' comments:

Reviewer's Responses to Questions

**Comments to the Author**

1. Is the manuscript technically sound, and do the data support the conclusions?

Reviewer #1: Partly

2. Has the statistical analysis been performed appropriately and rigorously? 

Reviewer #1: No

3. Have the authors made all data underlying the findings in their manuscript fully available?

Reviewer #1: No

4. Is the manuscript presented in an intelligible fashion and written in standard English?

Reviewer #1: No

5. Review Comments to the Author

Reviewer #1: This paper analyzes the diagnostic value and the complication rate of US-TCNB, particularly large bore cutting biopsy in patients with mediastinal lesions. The authors show that US-TCNB of mediastinal lesions is highly effective and safe tool which is

particularly helpful in critically ill patients. This article is not innovative. There are several issues that make this concept as currently presented unconvincing:

1. Inclusion criteria should be described in details. For example, why chose only dimension of the lesions more than 15 mm.

2. Characteristics of participants should be described, such as age, lesion location, size of lesion.

3. Study patients by diagnosis should be also described in details.

4. In lines 259-264, the result is inconvincible. The details of US-guided CNB including or excluding insufficient samples should be described. What about the positive predictive and the negative predictive?

5. The figure legends and tables should be moved to the end of the paper. Moreover, the tables is difficult to read and should be modified.

6. The manuscript needs more work on improving the English and the clarity and focus of the presented arguments.

6. PLOS authors have the option to publish the peer review history of their article (what does this mean?). If published, this will include your full peer review and any attached files.

Reviewer #1: No

---

## [Author Response · Author response to Decision Letter 0]

28 Jan 2020

Dear Dr. Alizad,

Thank you and the Reviewers for the constructive feedback provided. I had tried to address the pointed issues and feel the manuscript is significantly improved. I am uploading a revised version with revisions in “track changes”. I have responded to the reviewers’ comments below in bold. 

Kind regards,

Evgeni Mekov

Please ensure that your manuscript meets PLOS ONE's style requirements, including those for file naming. The PLOS ONE style templates can be found at http://www.plosone.org/attachments/PLOSOne_formatting_sample_main_body.pdf and http://www.plosone.org/attachments/PLOSOne_formatting_sample_title_authors_affiliations.pdf

We’ve tried to comply with all of the journals’ requirements. Please also note that the above links are not active. However, if there is still an issue we apologize and we are ready to resolve it upon request.

In the ethics statement in the manuscript and in the online submission form, please provide additional information about the patient records used in your retrospective study. Specifically, please ensure that you have discussed whether all data were fully anonymized before you accessed them. If patients provided informed written consent to have data from their medical records used in research, please include this information.

All the information is processed anonymously. The manuscript is edited to include this answer.

We noted in your submission details that a portion of your manuscript may have been presented or published elsewhere. ["Paper is containing original research and has not been submitted/published earlier in any journal and is not being considered for publication elsewhere. However there are 4 ABSTRACTS presented at the European Respiratory Congresses: ... They are not full text publications, nor is this manuscript submitted for consideration anywhere else. These are presentations of the work only in abstract form."] Please clarify whether this publication was peer-reviewed and formally published. If this work was previously peer-reviewed and published, in the cover letter please provide the reason that this work does not constitute dual publication and should be included in the current manuscript.

Several abstracts are presented on congresses. These include a part of the results. These papers are not peer-reviewed per se:

- Usually, they are reviewed by 1 person who rates them on a scale and based on the results they are accepted or rejected;

- Writing a review is not required for this type of evaluation;

- They are not handled by at least two peers, nor are they handled by an academic editor;

- We didn’t fill the copyright agreement form;

- Moreover, they are less than 400 words and are not considered a full-text publication. 

However, the ERS representative could give you more information if there is still an issue on this topic. As a rule, the majority of the researchers present their work at congresses and subsequently publish it as a full text.

We note that you have stated that you will provide repository information for your data at acceptance. Should your manuscript be accepted for publication, we will hold it until you provide the relevant accession numbers or DOIs necessary to access your data. If you wish to make changes to your Data Availability statement, please describe these changes in your cover letter and we will update your Data Availability statement to reflect the information you provide.

I confirm uploading raw data upon acceptance.

Reviewers' comments:

Reviewer #1: This paper analyzes the diagnostic value and the complication rate of US-TCNB, particularly large bore cutting biopsy in patients with mediastinal lesions. The authors show that US-TCNB of mediastinal lesions is highly effective and safe tool which is particularly helpful in critically ill patients. This article is not innovative. There are several issues that make this concept as currently presented unconvincing:

1. Inclusion criteria should be described in details. For example, why chose only dimension of the lesions more than 15 mm.

Lesions less than 15 mm are deemed unsuitable for US-TCNB because of increased risk of iatrogenic complications. The cutting diameter of this type of needles is 20 mm, therefore even 15 mm lesions possess a significant risk of complication even if they are placed at the optimal position because they will cut more than a maximal lesion diameter. However, we could manage this issue by cutting the tissues of the chest wall (e.g. muscle tissue) rather than lung parenchyma. We don’t have experience with lesions <15 mm, but we don’t recommend this technique in these cases.

The other criteria are self-explanatory, we need cooperative patients (some cases require holding breath, e.g. when the lesion is moving or behind the rib, etc), normal coagulation is required in any type of biopsy. Bronchoscopy is a standard diagnostic procedure and is performed in every patient with such lesions. Moreover, it is required preoperatively. If there are other queries or issues we could discuss them.

2. Characteristics of participants should be described, such as age, lesion location, size of lesion.

Lesion location and size are described in table 2. The age is described in the paper (page 9, row 171, 172).

3. Study patients by diagnosis should be also described in details.

Study patients by diagnosis are described in table 2.

4. In lines 259-264, the result is inconvincible. The details of US-guided CNB including or excluding insufficient samples should be described. What about the positive predictive and the negative predictive?

I am not sure I understand you properly what is your query. Insufficient samples by means of undiagnostic samples should be included as a part of false negatives. Otherwise, please specify.

5. The figure legends and tables should be moved to the end of the paper. Moreover, the tables is difficult to read and should be modified.

According to the submission guidelines, they should stay as is. This is an excerpt from https://journals.plos.org/plosone/s/submission-guidelines:

Other elements:

- Figure captions are inserted immediately after the first paragraph in which the figure is cited. Figure files are uploaded separately.

- Tables are inserted immediately after the first paragraph in which they are cited.

 Regarding the difficulty, could you please be more specific about this particular issue? What table could eventually imply difficulty to modify it for the readers?

6. The manuscript needs more work on improving the English and the clarity and focus of the presented arguments

The manuscript is edited as suggested

I would like to thank you and reviewers for the helpful feedback.

---

## [Decision Letter · Decision Letter 1]

26 Mar 2020

Diagnostic value and complication rate of ultrasound-guided transthoracic core needle biopsy in mediastinal lesions

PONE-D-19-21908R1

Dear Dr. Mekov,

We are pleased to inform you that your manuscript has been judged scientifically suitable for publication and will be formally accepted for publication once it complies with all outstanding technical requirements.

With kind regards,

Azra Alizad, MD

Academic Editor

PLOS ONE

Additional Editor Comments (optional):

Thank you for addressing the comments of previous reviews. It is a great work.

Reviewers' comments:

Reviewer's Responses to Questions

**Comments to the Author**

1. If the authors have adequately addressed your comments raised in a previous round of review and you feel that this manuscript is now acceptable for publication, you may indicate that here to bypass the “Comments to the Author” section, enter your conflict of interest statement in the “Confidential to Editor” section, and submit your "Accept" recommendation.

Reviewer #2: All comments have been addressed

2. Is the manuscript technically sound, and do the data support the conclusions?

Reviewer #2: Yes

3. Has the statistical analysis been performed appropriately and rigorously? 

Reviewer #2: Yes

4. Have the authors made all data underlying the findings in their manuscript fully available?

Reviewer #2: Yes

5. Is the manuscript presented in an intelligible fashion and written in standard English?

Reviewer #2: Yes

6. Review Comments to the Author

Reviewer #2: I have no additional comments. The authors have done a good job addressing the previous reviews. Thank you

7. PLOS authors have the option to publish the peer review history of their article (what does this mean?). If published, this will include your full peer review and any attached files.

Reviewer #2: No

---

## [Editor Report · Acceptance letter]

31 Mar 2020

PONE-D-19-21908R1 

Diagnostic value and complication rate of ultrasound-guided transthoracic core needle biopsy in mediastinal lesions 

Dear Dr. Mekov:

I am pleased to inform you that your manuscript has been deemed suitable for publication in PLOS ONE. Congratulations! Your manuscript is now with our production department. 

With kind regards,

on behalf of

Dr. Azra Alizad 

Academic Editor

PLOS ONE